# Thailand Raw Water Quality Dataset Analysis and Evaluation

Jaturapith Krohkaew [1], Pongpon Nilaphruek [1,*], Niti Witthayawiroj [2], Sakchai Uapipatanakul [3], Yamin Thwe [4] and Padma Nyoman Crisnapati [5]

1 Department of Big Data Management and Analytics, Rajamangala University of Technology Thanyaburi, Pathum Thani 12110, Thailand; kjatura@rmutt.ac.th
2 Department of Computer Science, Rajamangala University of Technology Thanyaburi, Pathum Thani 12110, Thailand; niti@rmutt.ac.th
3 Kinetics Corporation Ltd., 388 Ratchadapisek Rd.32 Chandrakasem, Chatuchak, Bangkok 10900, Thailand; sakchai@kinetics.co.th
4 Department of Data and Information Science, Rajamangala University of Technology Thanyaburi, Pathum Thani 12110, Thailand; yamin_t@mail.rmutt.ac.th
5 Department of Mechatronics Engineering, Rajamangala University of Technology Thanyaburi, Pathum Thani 12110, Thailand; padma_c@mail.rmutt.ac.th
* Correspondence: pongpon_n@rmutt.ac.th

**Abstract:** AbstractSustainable water quality data are important for understanding historical variability and trends in river regimes, as well as the impact of industrial waste on the health of aquatic ecosystems. Sustainable water management practices heavily depend on reliable and comprehensive data, prompting the need for accurate monitoring and assessment of water quality parameters. This research describes a reconstructed daily water quality dataset that complements rare historical observations for six station points along the Chao Phraya River in Thailand. Internet of Things technology and a Eureka water probe sensor is used to collect and reconstruct the water quality dataset for the period from June 2022–February 2023, with Turbidity, Optical Dissolved Oxygen, Dissolved Oxygen Saturation, Spatial Conductivity, Acidity/Basicity, Total Dissolved Solids, Salinity, Temperature, Chlorophyll, and Depth as the recorded parameters from six different stations. The presented dataset comprises a total of 211,322 data points, which are separated into six CSV files. The dataset is then evaluated using the Long Short-Term Memory (LSTM) algorithm with a Mean Squared Error (MSE) of 0.0012256, and Root Mean Squared Error (RMSE) of 0.0350080. The proposed dataset provides valuable insights for researchers studying river ecosystems, supporting informed decision-making and sustainable water management practices.

**Keywords:** water quality dataset; Internet of Things; real-time monitoring; metropolitan waterworks authority; Thailand

## 1. Summary

The assessment of environmental health can be accomplished by considering five key factors: soil, water, climate, natural vegetation, and landscapes. Out of these elements, water plays the most critical role in supporting human life and the survival of various ecosystems [1]. Its importance extends to drinking, household use, food production, and recreation, making safe and clean water an essential requirement for public health [2]. Therefore, it is vital to maintain proper water quality for preventing significant harm to human well-being and for maintaining an ecological balance for other species [3]. Water pollution, a significant global problem, requires ongoing evaluation and international efforts to effectively manage water resources, from a broader perspective to individual

wells. Numerous studies have shown that water pollution is a leading cause of death and illness worldwide, resulting in numerous daily fatalities [4]. In numerous developing nations, untreated or contaminated water is consumed due to public and administrative ignorance, coupled with the absence of a water quality monitoring system, leading to severe health complications [5,6]. Predicting and providing early warnings regarding pollution or declining water quality can serve as effective preventive measures that can be implemented promptly, especially in Thailand.

In Thailand, the Metropolitan Waterworks Authority (MWA) has the primary responsibility to supply and distribute water to various regions. They achieve this by mainly utilizing the raw water resources from the Chao Phraya River. Several research studies have been carried out in cooperation with the Metropolitan Waterworks Authority (MWA) on raw water quality management in the Chao Phraya River. Ref. [7] suggested innovative approaches for managing the saltwater influx at the Samlae Water Pumping Station, which serves as the primary intake station for the MWA (Metropolitan Waterworks Authority) from the Chao Phraya River. Additionally, they emphasized the need for enhanced cooperation from the MWA in water resource management efforts. Ref. [8] concluded that the urban water supply systems generally met water quality standards, except for color and iron issues caused by sedimentation process problems and iron pipe presence. Ref. [9] proposed the one-dimensional simulation flow model that proved valuable for optimizing water management, and enabling energy savings and efficient emergency water discharge planning in the West Water Canal. Ref. [10] identifies the optimal coagulants, their respective dosages, and cost efficiencies for effectively treating raw water with low, normal, and high turbidity levels, enabling the Metropolitan Waterworks Authority to meet water quality standards with greater clarity and cost-effectiveness. Based on the previous studies that have been conducted, the focus is only on developing methods for distributing and monitoring the water quality of the Chao Phraya River. However, a complete dataset concerning the water quality conditions of the river, especially in English, is still lacking. Therefore, there is a recognized need to provide comprehensive water quality datasets for researchers. The availability of such data will not only benefit researchers but also contribute to the sustainable development and protection of this vital natural resource in Thailand. To ensure the availability of this dataset in real time, which can be accessed anytime and anywhere, the selection of the right technology is very important.

This paper employs Internet of Things (IoT) technology to collect data, enabling the monitoring of water quality through sensors immersed in water. By employing diverse sensors, this system captures various essential parameters from the water. The rapid advancement of Wireless Sensor Network (WSN) technology has revolutionized real-time data acquisition, transmission, and processing, allowing users to access sustainable water quality information remotely. IoT has become a groundbreaking phenomenon with applications spanning various fields, including smart cities, smart power grids, smart supply chains, and smart wearables [11]. Although IoT is yet to reach its full potential in environmental applications, it offers immense opportunities. It can be utilized for detecting forest fires and early earthquakes, reducing air pollution, monitoring green houses, preventing landslides, and most importantly, for water quality monitoring and control systems [12–20]. In the twenty-first century, researchers have placed considerable emphasis on monitoring water quality, leading to numerous ongoing projects that explore various aspects of this field. The main objective of these research is to create a monitoring system for water quality that is both efficient and cost-effective, while also providing real-time data. This system would integrate wireless sensor networks and the Internet of Things (IoT), enabling comprehensive monitoring of water quality parameters [21]. In addition to monitoring systems, another crucial focus is ensuring the availability of datasets for researchers. These datasets are essential for developing artificial intelligence models aimed at predicting and preventing disasters related to water quality.

Therefore, in this study, a dataset comprising daily river water quality measurements collected from six stations along the Chao Phraya River in Thailand was presented. The

dataset was obtained using IoT technology, specifically the Eureka Water Probe Manta +35 sensors deployed at each station, enabling accurate real-time monitoring of river water conditions. The Eureka Manta water quality multiprobe underwent rigorous testing at the U.S. Geological Survey (USGS) Hydrologic Instrumentation Facility to assess its accuracy and compliance with standards, including ISO 7027 [22] for measuring turbidity and Standard Methods 2510 B to correct the specific conductance. The results demonstrated that the Manta met the criteria outlined in the USGS National Field Manual for continuous water quality monitors, covering parameters such as dissolved oxygen and turbidity.

The sensor measurements encompass parameters such as Turbidity (TURB_NTU), Optical Dissolved Oxygen (HDO), Dissolved Oxygen Saturation (DO_SAT), Spatial Conductivity (SPCOND), Acidity/Basicity (pH), Total Dissolved Solids (TDS), Salinity (SALINITY), Temperature (TEMP), Chlorophyll (CHL), and Depth (DEPTH). Water conditions are recorded and stored in a MySQL database at 10-min intervals. The available dataset can serve a wide range of applications, encompassing a trend analysis, heat flux calculations, calibration/validation of water temperature models based on processes, establishing baseline conditions for future climate projects, analyzing climate drivers, assessing impacts on ecosystem health, and evaluating water quality.

Previous studies have collected river water quality data with varying recording parameters, timings, and locations as summarized in Table 1, which highlights the importance of conducting this research update. Hence, the specific objectives of this study are as follows: (1) collecting a novel dataset on water quality in Thailand, specifically along six points of the Chao Phraya River, and (2) employing the Long Short-Term Memory (LSTM) model, in order to evaluate the quality of our proposed dataset.

**Table 1.** Previous river water quality dataset.

| Title | Targeted Domain | Data Range (Time) | Place |
|---|---|---|---|
| Dataset on the assessment of water quality and water quality index of Ubogo and Egini rivers, Udu LGA, Delta State Nigeria [23] | PH value, Total Dissolved Solid, Water Temperature, variations in Conductivity, Dissolved Oxygen Values, Chloride, Biochemical Oxygen Demands, Turbidity, Hardness, Nitrate, Sulphate, Phosphate, Calcium, Magnesium value, Potassium. | February–July 2010 | Ubogo and Egini Rivers, Nigeria |
| Ref. [23]: the dataset focuses on evaluating the water quality of surface water in the Kalingarayan Canal, specifically concerning heavy metal pollution in the Tamil Nadu region [24] | The dataset includes measurements of eight specific heavy metals, namely iron, copper, manganese, chromium, zinc, cadmium, lead, and nickel. | January 2014–December 2016 | Kalingarayan Canal, Tamil Nadu |
| FIKWater: A Water Consumption Dataset from Three Restaurant Kitchens in Portugal [25] | Hot and cold water demand. | February–May 2019 | Restaurant kitchens in Portugal |
| Datasets of Groundwater Level and Surface Water Budget in a Central Mediterranean Site [26] | Hydrological and atmospheric variables. | 21 June 2017–1 October 2022 | Central Mediterranean, the Salento Peninsula |
| Reconstructed River Water Temperature Dataset for Western Canada [27] | River water temperature. | 1980–2018 | 55 river stations across western Canada |
| Raw Water Quality Dataset (Proposed Dataset) | Turbidity, Optical Dissolved Oxygen, Dissolved Oxygen Saturation, Spatial Conductivity, pH, Total Dissolved Solids, Salinity, Temperature, Chlorophyll, and Depth. | June 2022–February 2023 | Six stations across the Chao Phraya River, Thailand |

## 2. Dataset Description

The collected dataset contains various information such as error logs, wipe schedules, and sensor logs, therefore filtering data is carried out first to separate sensor logs from other data. The data obtained from the database are in the form of MySQL (.sql) files that are then filtered and converted into Comma Separated Values (.csv), which separates data between stations. The naming of "XX Logs.csv" was performed to make it easier to categorize by station, where XX is the station ID. There are six CSV dataset files named s1 Logs.csv, s2 Logs.csv, s3 Logs.csv, s4 Logs.csv, s5 Logs.csv, and s15 Logs.csv. In the dataset, the comma symbol (,) is employed as a separator between columns, while the dot symbol (.) is utilized to indicate decimal values. The initial row of the CSV file includes the titles for each data column, which can be observed in Figure 1. Additionally, the distribution of our dataset for station 1 is illustrated in Figure 2a–j. The turbidity (NTU) of water should be lower for better clarity, while a higher optical dissolved oxygen (HDO) level is desirable. Lower values of Spatial Conductivity (SPCOND) indicate less saltiness, and the pH range of water should ideally be between 6.5 and 8.5. Lower total dissolved solids (TDS) below 1000 are preferable. Salinity represents the dissolved salt content of a body of water, and the temperature typically falls within the range of 43 to 68 degrees Fahrenheit.

| id | station_id | date | time | turb_ntu | hdo | hdo_sat | spcond | ph | tds | salinity | temp | chl | depth | created_at | update_at |
|---|---|---|---|---|---|---|---|---|---|---|---|---|---|---|---|
| 1 | S1 | 2022-06-14 | 13:40:00 | 65.52 | 2.11 | 29.7 | 419 | 7.36 | 268.7 | 0.20 | 31.89 | 2.77 | 0.20 | 2022-06-14 06:49:24 | 2022-06-18 15:09:40 |
| 2 | S1 | 2022-06-14 | 13:50:00 | 43.96 | 2.09 | 29.5 | 419 | 7.36 | 268.4 | 0.20 | 31.90 | 2.41 | 0.20 | 2022-06-14 06:52:25 | 2022-06-18 15:09:41 |
| 3 | S1 | 2022-06-14 | 14:00:00 | 36.12 | 2.06 | 29.0 | 421 | 7.36 | 269.8 | 0.20 | 31.91 | 2.20 | 0.20 | 2022-06-14 07:01:25 | 2022-06-18 15:09:41 |
| 4 | S1 | 2022-06-14 | 14:10:00 | 44.62 | 2.04 | 28.8 | 430 | 7.34 | 275.2 | 0.20 | 31.92 | 2.82 | 0.20 | 2022-06-14 07:10:25 | 2022-06-18 15:09:41 |
| 5 | S1 | 2022-06-14 | 14:20:00 | 38.01 | 2.09 | 29.4 | 421 | 7.36 | 269.4 | 0.20 | 31.93 | 2.27 | 0.20 | 2022-06-14 07:22:27 | 2022-06-18 15:09:41 |
| ... | ... | ... | ... | ... | ... | ... | ... | ... | ... | ... | ... | ... | ... | ... | ... |
| 35314 | S1 | 2023-02-15 | 06:40:00 | 12.38 | 3.77 | 48.4 | 361 | 7.23 | 231.1 | 0.17 | 28.03 | 3.15 | 1.83 | 2022-09-12 22:10:48 | 2023-02-14 23:40:17 |
| 35315 | S1 | 2023-02-15 | 06:50:00 | 12.06 | 3.76 | 48.3 | 361 | 7.24 | 231.2 | 0.17 | 28.02 | 3.14 | 1.84 | 2022-09-12 22:10:48 | 2023-02-14 23:50:16 |
| 35316 | S1 | 2023-02-15 | 07:00:00 | 11.47 | 3.75 | 48.2 | 361 | 7.23 | 231.3 | 0.17 | 28.03 | 3.44 | 1.85 | 2022-09-12 22:10:48 | 2023-02-15 00:00:17 |
| 35317 | S1 | 2023-02-15 | 07:10:00 | 13.16 | 3.75 | 48.1 | 361 | 7.23 | 231.4 | 0.17 | 28.03 | 3.09 | 1.85 | 2022-09-12 22:10:48 | 2023-02-15 00:10:17 |
| 35318 | S1 | 2023-02-15 | 07:20:00 | 12.35 | 3.73 | 47.9 | 361 | 7.23 | 231.5 | 0.17 | 28.03 | 3.34 | 1.87 | 2022-09-12 22:10:48 | 2023-02-15 00:20:17 |

**Figure 1.** CSV format dataset sample.

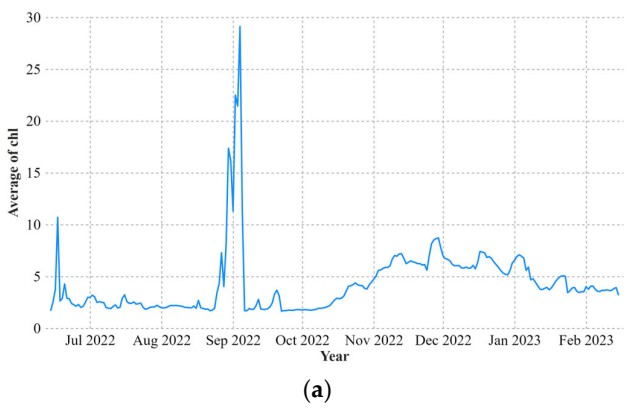

(**a**)

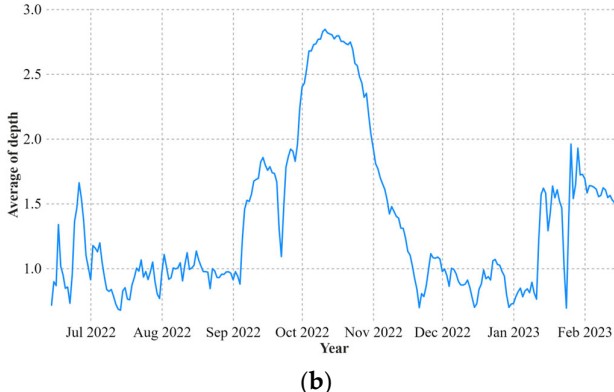

(**b**)

**Figure 2.** *Cont.*

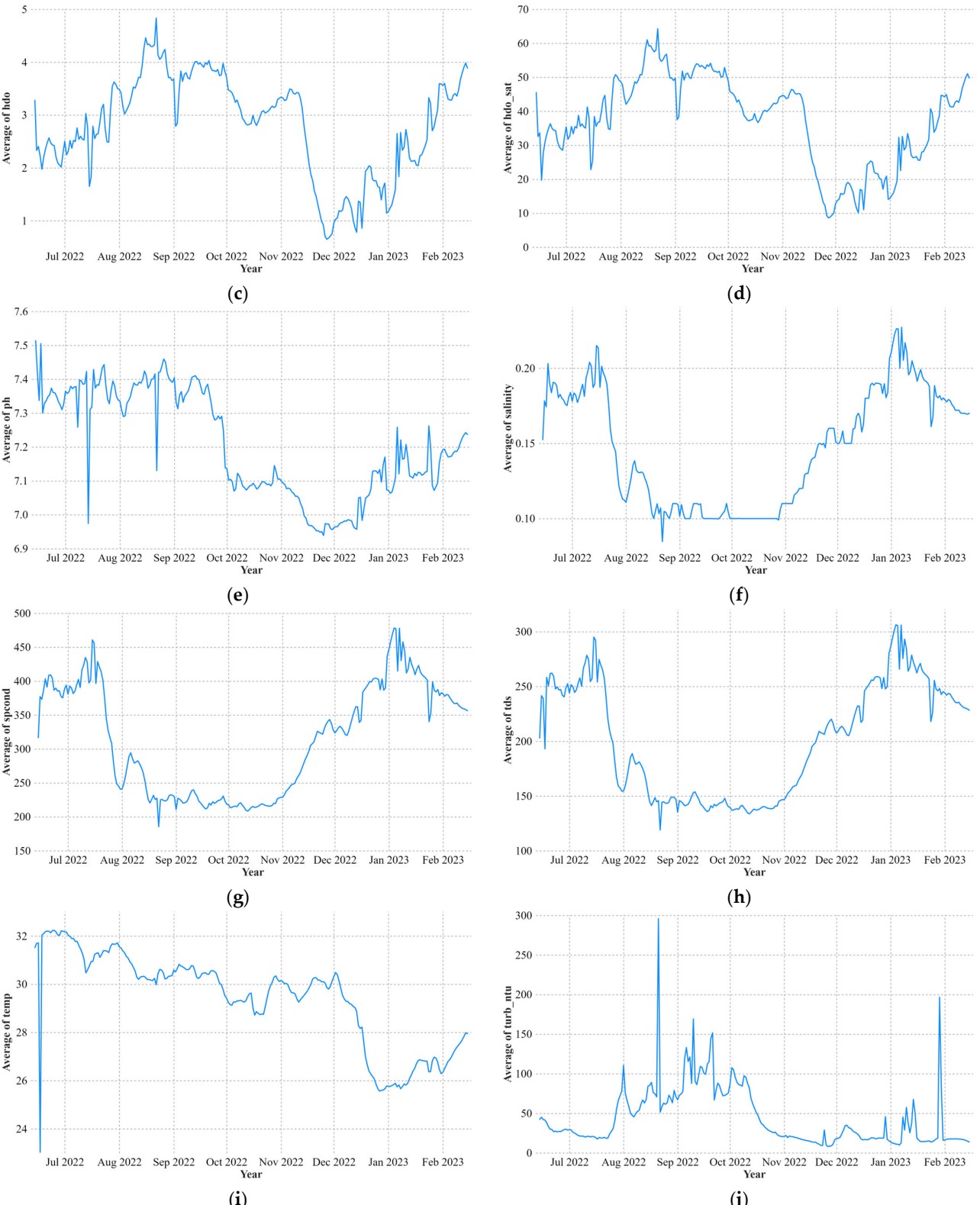

**Figure 2.** Average data distribution chart by date for station 1. (**a**) Chlorophyll; (**b**) Depth; (**c**) Dissolved Oxygen; (**d**) Dissolved Oxygen Saturation; (**e**) pH; (**f**) Salinity; (**g**) Spatial Conductivity; (**h**) Total Dissolved Solids; (**i**) Temperature; (**j**) Turbidity.

The dataset consists of 16 columns with ID values as the primary one and station_id as markers for each station. Table 2 describes each of the water quality parameters' data

collected from sensors at each station. The dataset obtained still contains noise in the form of lost values due to disconnected internet connections, therefore data cleansing is carried out using formula (1) where $d_i$ is noise data, and $_{i-1}$ and $_{i+1}$ correspond to the previous and next valid measurements relative to the missing data point $_i$.

$$d_i = \frac{d_{i-1} + d_{i+1}}{2} \tag{1}$$

**Table 2.** Dataset column name and type.

| No. | Column | Data Type | Description | Data Range |
|---|---|---|---|---|
| 1 | id | int64 | Record ID | [0, 35,316) for S1, [0, 35,741) for S2, [0, 34,612) for S3, [0, 35,185) for S4, [0, 34,691) for S5 and [0, 35,777) for S15 |
| 2 | station_id | object | Station ID | 'S1', 'S2', 'S3', 'S4', 'S5', 'S15' |
| 3 | date | datetime64[ns] | Local time YYYY:MM:DD | June 2022 to February 2023 |
| 4 | time | object | Local time HH:MM:SS | From 12:00 AM until 11:59 PM |
| 5 | turb_ntu | float64 | Turbidity value | 0 to 1000 FNU |
| 6 | hdo | float64 | Optical Dissolved Oxygen value | 0 to 20 mg/L |
| 7 | hdo_sat | float64 | Dissolved Oxygen Saturation value | 0 to 500% saturation |
| 8 | spcond | int64 | Spatial Conductivity value | 0 to 5000 µS/cm |
| 9 | ph | float64 | Acidic/Basic value | 0 to 14 units |
| 10 | tds | float64 | Total Dissolved Solids value | 0 to 65 g/L |
| 11 | salinity | float64 | Salinity value | 0 to 70 PSU |
| 12 | temp | float64 | Temperature value | −5 to 50 °C |
| 13 | chl | float64 | Chlorophyll value | 0 to 100 µg/L |
| 14 | depth | float64 | Depth value | 0 to 3.25 m |
| 15 | created_at | object | Local time YYYY:MM:DD: HH:MM:SS | - |
| 16 | update_at | object | Local time YYYY:MM:DD: HH:MM:SS | - |

Data preprocessing holds significant importance within the data analysis and machine learning pipeline. It encompasses the identification and rectification of errors, inconsistencies, and inaccuracies in a dataset to enhance its quality and reliability. In the current scenario, the provided datasets were collected from six distinct water stations, which has introduced inconsistencies in the data formats. To address the issue of missing data, the standard data range was outlined in Table 2. This step ensures that the dataset remains consistent and reliable for further analysis.

Table 3 shows the distribution of the sum, mean, standard deviation, minimum value, and maximum value for Station 1 after the data preprocessing step. The correlation between the collected sensor parameters is presented in Figure 3. One important relationship is between hdo_sat and hdo, which demonstrates a close correlation because the value of dissolved oxygen in units of mg/l is converted to a percentage (%) referred to as dissolved oxygen saturation. Additionally, spcond exhibits a close correlation with both tds and salinity, indicating their interdependence. Interestingly, tds, salinity, and spcond are negatively correlated with both hdo and hdo_sat, suggesting that an increase in these variables may result in a decrease in water quality. On the other hand, variables such as turb_ntu, pH, chl, and temp exhibit low to medium correlations with other variables, implying that their impact on water quality might be more nuanced and influenced by additional factors. Understanding these interconnections aids in comprehending the complex dynamics of water quality assessment and management. For more details, Figure 4 presents a graph showing the correlation between the Spatial Conductivity, TDS, and Salinity parameters, as these three variables influence each other. Figure 5 displays the correlation between HDO and HDO Saturation values, where HDO influences HDO Saturation.

**Table 3.** Dataset range value.

| | turb_ntu | HDO | HDO_sat | Spcond | pH | TDS | Salinity | Temp | chl | Depth |
|---|---|---|---|---|---|---|---|---|---|---|
| Count | 35,316 | 35,316 | 35,316 | 35,316 | 35,316 | 35,316 | 35,316 | 35,316 | 35,316 | 35,316 |
| Mean | 43.176283 | 2.79217 | 37.10983 | 314.4010 | 7.20433 | 201.2760 | 0.14704 | 29.42158 | 4.30925 | 1.35832 |
| Std | 45.392016 | 1.07877 | 14.47037 | 90.41907 | 0.24021 | 58.11721 | 0.04287 | 2.145848 | 4.00676 | 0.65397 |
| Min | 0.000000 | 0.00000 | 0.000000 | 0.00000 | 0.00000 | 0.000000 | 0.00000 | 0.010000 | 0.00000 | 0.00000 |
| 25% | 17.470000 | 2.09000 | 26.60000 | 227.0000 | 7.08000 | 145.7000 | 0.11000 | 28.56000 | 2.02000 | 0.93000 |
| 50% | 25.870000 | 2.90000 | 38.70000 | 322.0000 | 7.16000 | 206.4000 | 0.15000 | 30.05000 | 3.37000 | 1.21000 |
| 75% | 64.295000 | 3.57000 | 47.40000 | 395.0000 | 7.36000 | 252.8000 | 0.18000 | 30.67000 | 5.78000 | 1.75000 |
| Max | 1000.0000 | 8.66000 | 106.0000 | 503.0000 | 10.7300 | 322.5000 | 0.40000 | 37.44000 | 86.0500 | 3.25000 |

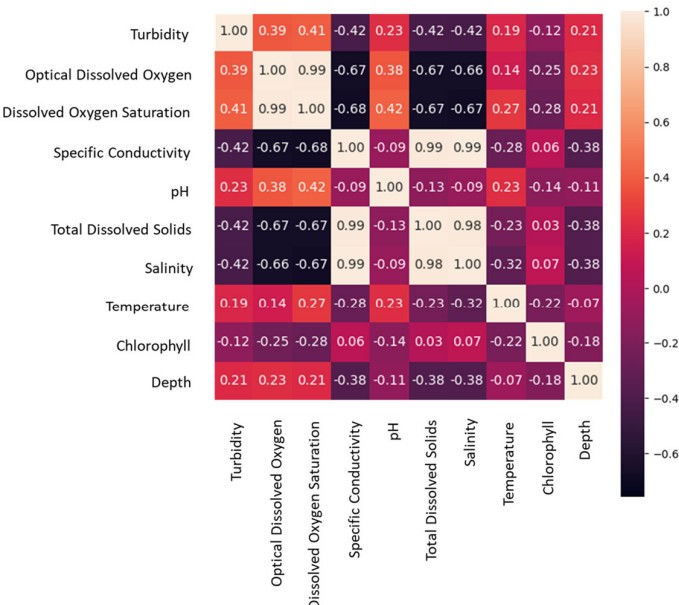

**Figure 3.** Data frame correlation matrix.

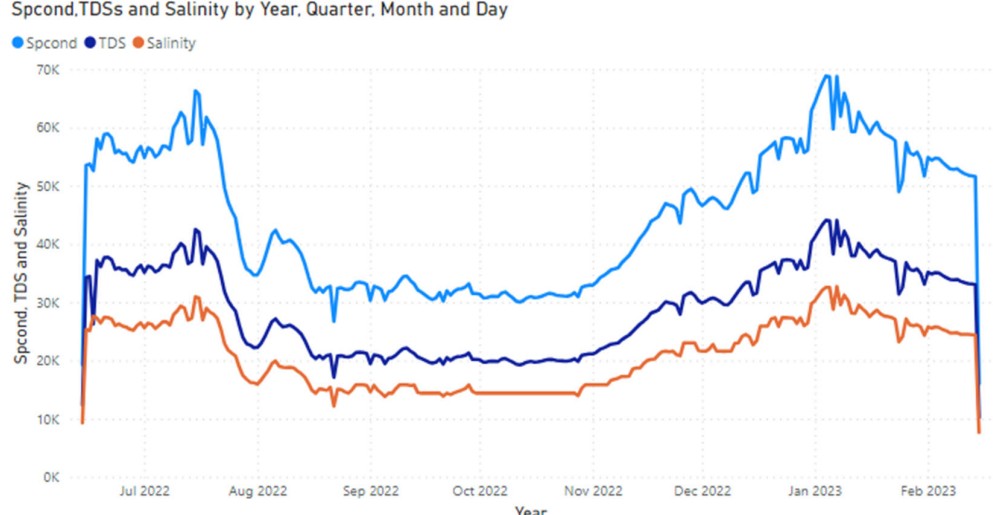

**Figure 4.** Correlation between spatial conductivity, TDS, and salinity.

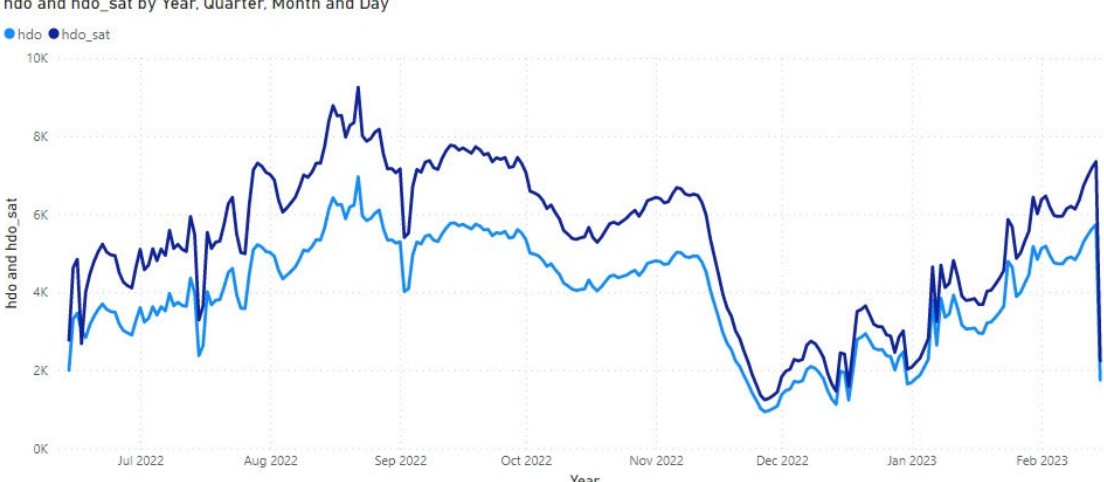

**Figure 5.** Correlation between HDO and HDO saturation.

## 3. Methods

This research selected six stations in Thailand based on the geographical location from upstream to downstream. In terms of implementation, a total of twenty-two stations have been strategically deployed along the Chao Phraya River. However, this dataset presents data on six stations that have obtained permission from the authorities. The location of each station is shown in Table 4. Figure 6a shows the specific location of the station installation in central Thailand. Figure 6b presents the distribution of stations along the river. Figure 6c–h provides detailed information on the location of the six stations from the satellite imagery.

**Table 4.** Locations of six stations.

| Station Name | Location | Longitude | Latitude |
|---|---|---|---|
| S1 | Sam Lae | 14.040804860627668 | 100.55605001072698 |
| S2 | Rangsit Siphon | 13.973218413649697 | 100.57142462277918 |
| S3 | Wat Phai Lom | 14.07795702917611 | 100.5258774272095 |
| S4 | Wat Makham | 14.004070095419458 | 100.540581425073 |
| S5 | Wat Pho Taeng Nuea | 14.131251392764735 | 100.52449883577663 |
| S15 | Bangkhen Water Treatment Plant | 13.8840299 | 100.5524927 |

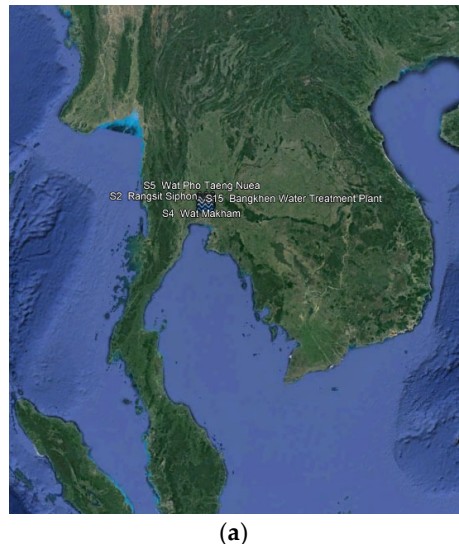

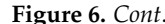

(**a**)

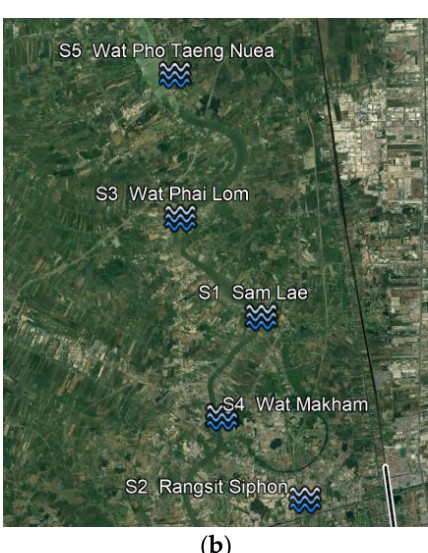

(**b**)

**Figure 6.** *Cont.*

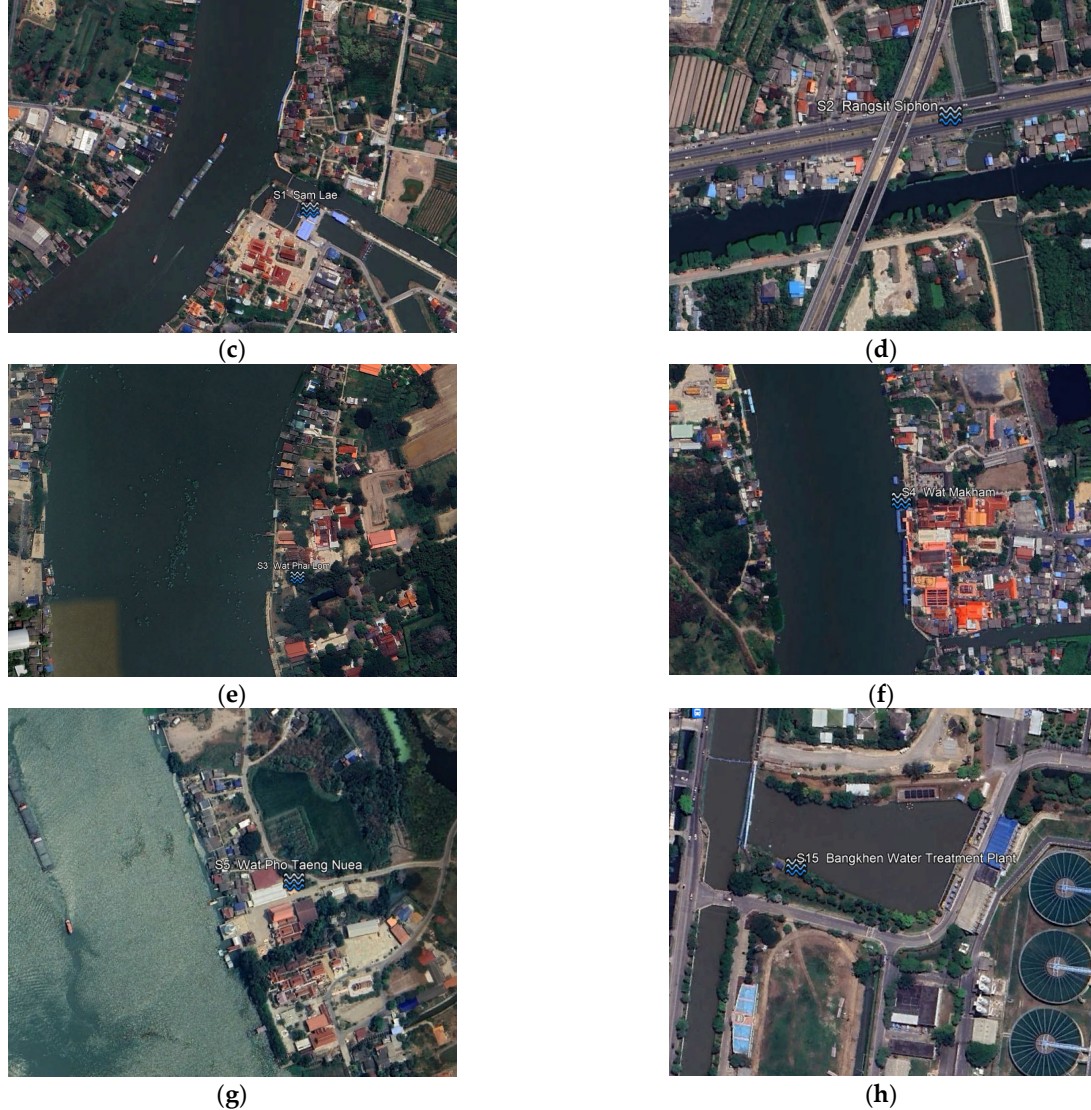

(c)  (d)

(e)  (f)

(g)  (h)

**Figure 6.** Six stations from the satellite. (**a**) Stations' locations based on Thailand map; (**b**) detailed location of each station; (**c**) Station 1; (**d**) Station 2; (**e**) Station 3; (**f**) Station 4; (**g**) Station 5; (**h**) Station 15.

### 3.1. Hardware Specification

The Eureka Manta +35 (Austin, TX, USA) multiprobe sensor (Figure 7) is used as a sensor to retrieve water quality data. Data from the sensors are then collected using a Mini PC K6-F13D (Bangkok, Thailand) as an IoT Gateway and are stored in a database. Figure 8 shows the deployment of one of the stations. To maintain the quality of the raw data collected, the Metropolitan Waterworks Authority of Thailand conducts a sensor calibration process every once per month (Figure 9).

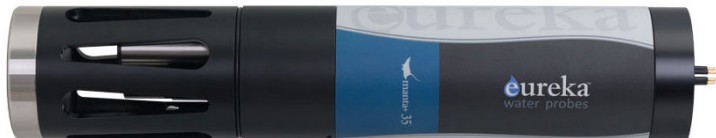

**Figure 7.** Eureka water probe.

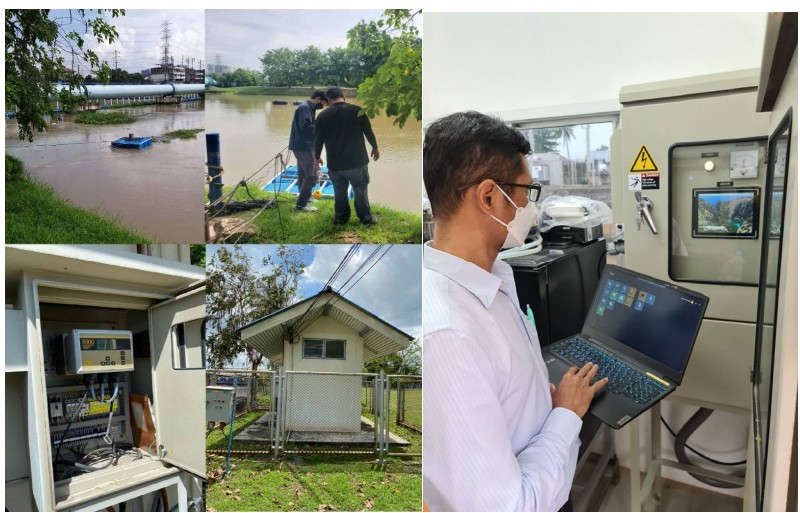

**Figure 8.** IoT hardware deployment.

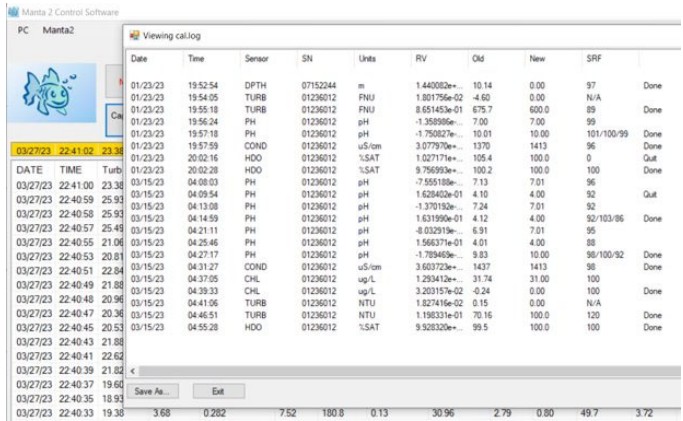

**Figure 9.** Sensor calibration activity.

### 3.2. System Overview

The data obtained are then stored in a database and displayed on http://rwc.mwa.co.th/page/info/ (accessed on 1 March 2023), which is a platform for displaying water quality data. The overview of the monitoring system used in collecting data is depicted in Figure 10. This system is divided into three main parts, namely Local Area Network (LAN), Cloud Server, and Web-Based data visualization. For the LAN section, the sensor is read in Python and then connected to the IoT gateway via a Serial Communication protocol. Furthermore, the data that have been obtained at each station are connected to the cloud server (MySQL) with the HTTP protocol. The data that have been collected can be accessed by the public on web-based applications.

### 3.3. Neural Network

The concept of a Neural Network (NN) is an imitation of the structure of the human brain's neural network. Neural networks, a fundamental component of deep learning, consist of multiple layers that work in harmony to process and analyze data. The key layers in a neural network include the input layer, hidden layer(s), and output layer. The input layer receives the initial data and serves as the network's entry point. Hidden layers, positioned between the input and output layers, extract meaningful patterns and representations from the input. Finally, the output layer provides the final predictions or results of the network's computations.

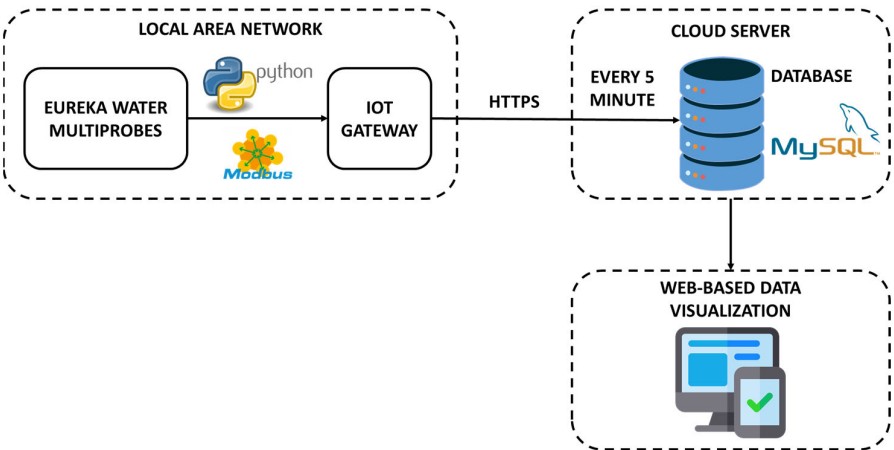

**Figure 10.** Real-time monitoring system overview.

Neural networks utilize forward propagation to generate predictions by passing data through the layers, and backward propagation to adjust parameters based on prediction errors during training [28]. Many types of neural network algorithms are well known in the world. Neural networks use interconnected layers for processing data, with forward propagation generating predictions and backward propagation adjusting parameters to minimize errors. This iterative process enables accurate predictions and the learning of complex relationships in various tasks. Figure 11 shows the commonly used three-layer neural network structure, consisting of input, hidden, and output layers [29].

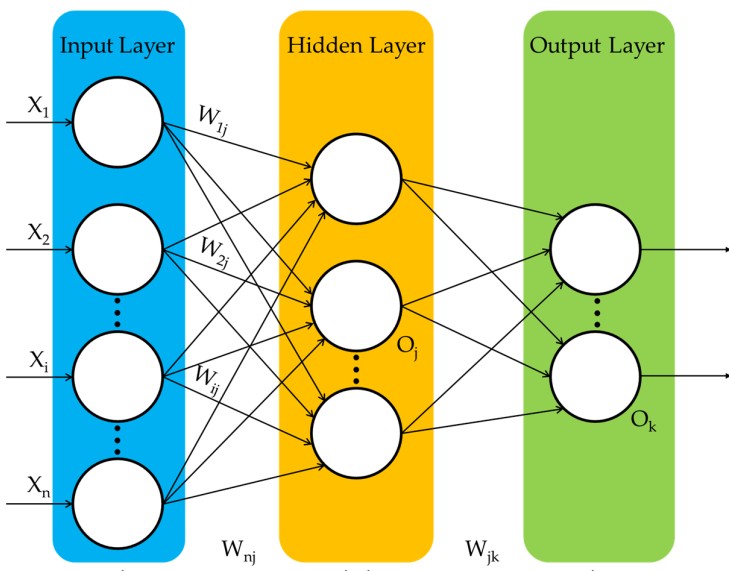

**Figure 11.** Three-layer neural network structure diagram.

The utilization of Backpropagation (BP) in neural network architectures involves various specific procedures, beginning with the initialization of weights and biases within the range of $-1$ to 1. Subsequently, the input value for each node in the hidden layer is calculated, as depicted in Equation (1). Equation (2) is then used to compute the output value for each node in the hidden layer. Equations (1) and (2) are reused to calculate input and output values at the output layer. The error value that occurs between the predicted and real values is calculated using Equation (3) and then BP is carried out. The error value in the hidden layer is calculated using Equation (4), then an update is made of the weight between each nerve node with Equations (5) and (6). Apart from the weight, the bias also needs to be updated using Equations (7) and (8). The process of calculating

Equations (2)–(11) continues to be carried out sequentially until the conditions are met. Table 5 provides the symbols' descriptions for equations used in this article.

$$I_j = \sum_i W_{ij}O_i + \theta_j \tag{2}$$

$$O_i = \frac{1}{1 + e^{-I_j}} \tag{3}$$

$$Err_j = O_j(1 - O_j)(T_j - O_j) \tag{4}$$

$$Err_j = O_j(1 - O_j)\sum_k Err_k W_{jk} \tag{5}$$

$$\Delta W_{ij} = (l)Err_j O_i \tag{6}$$

$$W_{ij} = W_{ij} + \Delta W_{ij} \tag{7}$$

$$\Delta\theta_j = (l)Err_j \tag{8}$$

$$\theta_j = \theta_j + \Delta\theta_j \tag{9}$$

**Table 5.** Nomenclature.

| | | | |
|---|---|---|---|
| $W_{ij}$ | the weight attributed to the upper layer node | $h_{t-1}$ | the final LSTM output value at time $t-1$ |
| $\theta_j$ | the bias of the node | $x_t$ | the input at time $t$ |
| $I_j$ | signifies the value inputted into the neural node | $\sigma$ | the activation function of sigmoid |
| $O_i$ | the output value of the upper layer of the node | $f_t$ | the output of input gate at time $t$ |
| $O_j$ | denotes the resulting value of the node's output | $i_t$ | denotes the output value produced by the input gate during time $t$ |
| $T_j$ | represents the actual value of the node | $o_t$ | the output of the output gate at time $t$ |
| $Err_j$ | the error value of the node | $\tilde{C}_t$ | the candidate cell state at time $t$ |
| $Err_k$ | denotes the error value of the node's connected neural unit in the output layer | $C_t$ | refers to the state of the cell during time $t$ |
| $W_{jk}$ | the output layer weight connected by the node | $W_f$ | the weight of the forget gate |
| $\Delta W_{ij}$ | the value of weight change | $W_i$ | the weight of the input gate |
| $\Delta\theta_j$ | change in the bias value | $W_o$ | the weight of the output gate |
| $l$ | the learning rate of the neural network | $W_c$ | the weight of the candidate input gate |
| $C_{t-1}$ | the cell state at time $t-1$ | $h_t$ | the cell state at time $t$ |
| $b_f$ | the weight of the forget gate | $b_c$ | the bias of the candidate input gate |
| $b_i$ | the weight of the input gate | $b_o$ | the bias of the candidate output gate |
| $h_t$ | the final output value | | |

## 4. Dataset Experiments and Evaluation

In this study, LSTM is used to evaluate water quality datasets, especially those involving time series data. LSTM is an algorithm developed from the Recurrent Neural Network (RNN), and this algorithm is designed based on traditional RNN problems related to explosions and the loss of gradients from data stored for a long time [30]. The significant difference seen in the standard RNN structure with the LSTM is the number of repeating modules. Standard RNN has a simple structure, for example, RNN only has one tanh layer, whereas LSTM has more than one tanh layer and they interact in a unique way [30]. Figure 12 shows the three main parts of the LSTM architecture, namely Forget, Input, and

Output Gate (FG, IG, OG). In calculations (10)–(15) it can be seen that $h_{t-1}$ (which is output) and $x_t$ (which is input) are inputs from FG, IG, Cell Update, and OG at time $t$.

$$f_t = \sigma\left(W_f[h_{t-1}, x_t] + b_f\right) \tag{10}$$

$$i_t = \sigma(W_i[h_{t-1}, x_t] + b_i) \tag{11}$$

$$\widetilde{C}_t = tanh(W_c[h_{t-1}, x_t] + b_c) \tag{12}$$

$$C_t = f_t \times C_{t-1} + i_t \times \widetilde{C}_t \tag{13}$$

$$o_t = \sigma(W_o[h_{t-1}, x_t] + b_0) \tag{14}$$

$$h_t = o_t \times tanh(C_t) \tag{15}$$

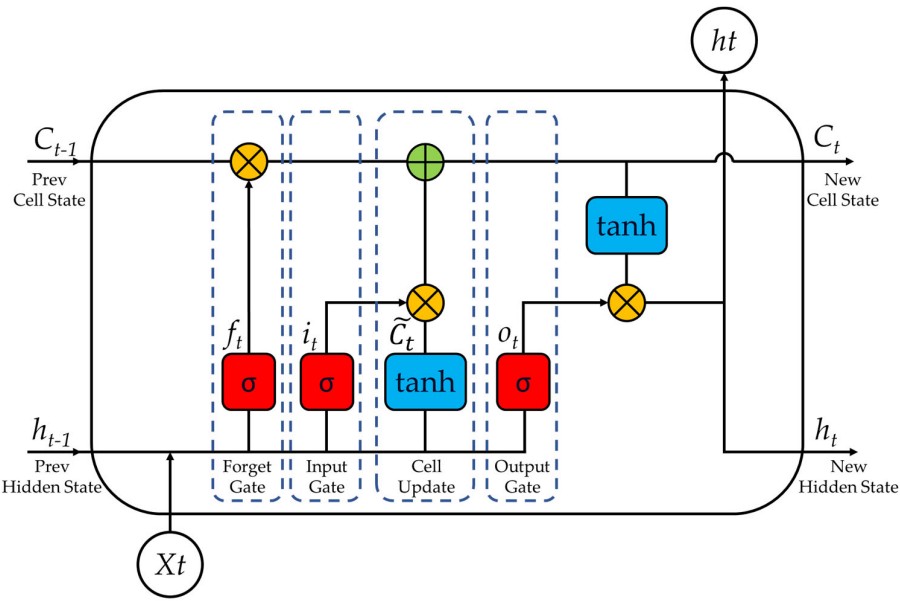

**Figure 12.** LSTM architecture.

LSTM's ability to capture long-term dependencies and handle sequential data makes it suitable for analyzing and predicting water quality parameters over time. The dataset used has gone through the process of data preparation and pre-processing. At this experiment and evaluation stage, data obtained at the S1-Sam Lae station were used. The dataset is then divided into two for training and testing purposes as shown in Figure 13. The training dataset spanned from July 2022 to December 2022 (75%), while the testing dataset covered the period from January 2023 to February 2023 (25%).

The parameter settings used in this study can be seen in Table 6, where Adam is used as the optimizer algorithm. We conducted an assessment of the LSTM model's predictive capabilities over a 45-day horizon, revealing its accurate prediction of a 10-day span. This outcome precisely corresponds to a calculated accuracy of 22.2%. In this scenario, the error rate is exceptionally high, leading to a correspondingly low level of accuracy achieved, which is because fine-tuning of the model has not been carried out yet. Evaluation of the LSTM model trained using test data is carried out by calculating performance metrics such as mean squared error (MSE) and root mean squared error (RMSE) to assess model accuracy in predicting water quality parameters. Statistical results for evaluating turbidity predictions can be seen in Figure 14 and Table 7. The MSE and RMSE metrics are in

common use and are especially suitable when the underlying data distribution follows a Gaussian behavior assuming normality of the data in this research. While the choice of MSE and RMSE is reasonable based on Gaussian assumptions, it is important to recognize that real-world datasets may exhibit deviations from this ideal distribution. Ref. [31] provides illustrative examples of situations where the data behavior deviates from normality. This reference highlights the importance of considering non-Gaussian behavior in practical applications, particularly in the context of water analysis and risk assessment. This can be material for further research for other researchers.

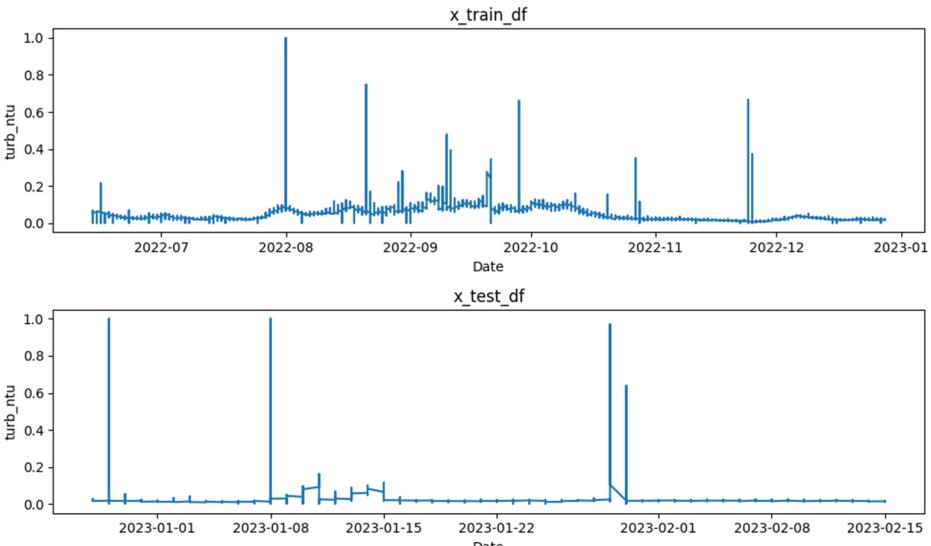

**Figure 13.** Training and testing dataset.

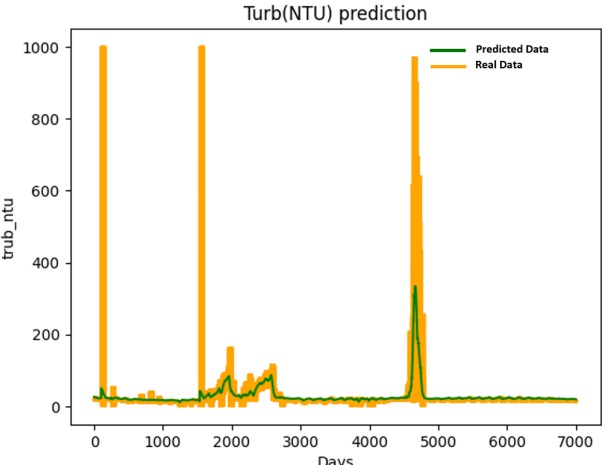

**Figure 14.** Experimental result of LSTM in predicting turbidity value.

**Table 6.** LSTM tuning parameters setting.

| Parameters | Value |
|---|---|
| This transfers function information from the input layer to the hidden layer | Sigmoid |
| The function responsible for activating the neural network | Tanh |
| The function used for optimizing the neural network | Adam |
| The count of elements in the input layer | 1 |
| The number of neurons in the hidden layer | 64 |
| The count of elements in the output layer | 1 |
| The size of each batch used for training | 32 |
| The time step used in the neural network model | 60 |
| The rate at which the neural network learns and adjusts its weights | 0.001 |

**Table 7.** LSTM prediction statistic results.

| Statistic | Value |
| --- | --- |
| Mean Squared Error (MSE) | 0.0012256 |
| Root Mean Squared Error (RMSE) | 0.0350080 |

## 5. Conclusions

In this study, a novel dataset has been successfully gathered, comprising observations from six strategically positioned stations along the Chao Phraya River in Thailand. To obtain a comprehensive understanding of the river's water quality conditions, ten key parameters were recorded using sensors. These parameters include Turbidity, Optical Dissolved Oxygen, Dissolved Oxygen Saturation, Spatial Conductivity, pH level, Total Dissolved Solids, Salinity, Temperature, Chlorophyll, and Depth. Water clarity is enhanced when turbidity (NTU) levels are lower, and higher optical dissolved oxygen (HDO) levels are preferable. Lower conductivity (SPCOND) values indicate reduced saltiness and maintaining a pH range between 6.5 and 8.5 is ideal. Moreover, lower total dissolved solids (TDS) below 1000 are preferable, while salinity reflects the dissolved salt content of water. Additionally, the temperature typically falls within the range of 43 to 68 degrees Fahrenheit. Based on the dataset distribution that has been collected, water quality standards were found to be met in some parameters of the Chao Phraya River, including HDO, pH, and TDS.

There are also correlations between Spatial Conductivity, TDS, and Salinity parameters, and they influence each other. Similarly, the relationship between HDO and HDO Saturation values indicates that these parameters are also influencing each other. For a comprehensive understanding, further exploration of these correlations is recommended. After the data collection phase, data preprocessing and evaluation has been performed on the dataset. Based on the parameters observed in the proposed dataset, it can be seen that the quality of water along the Chao Phraya River is good between August and November, likely due to the rainy season in Thailand. During the evaluation process, a deep learning LSTM model was employed, which exhibited suboptimal accuracy in predicting water quality. However, this dataset holds immense potential as a valuable resource for future research endeavors focused on monitoring water quality and establishing early warning systems for pollution-related disasters in Thailand. The insights from this study provide a foundation for advancing our understanding and management of water quality in the region.

**Author Contributions:** Conceptualization, J.K. and P.N.; methodology, J.K. and P.N.; software, J.K. and P.N.; validation, J.K. and P.N.; formal analysis, P.N.; investigation, J.K. and P.N.; resources, J.K. and P.N.; data curation, J.K. and P.N.; writing—original draft preparation, Y.T. and P.N.C.; writing—review and editing, J.K. and P.N.; visualization, J.K. and P.N.; supervision, J.K. and P.N.; project administration, J.K. and P.N.; funding acquisition, N.W. and S.U.; All authors have read and agreed to the published version of the manuscript.

**Funding:** This research was funded by Metropolitan Waterworks Authority Thailand, grant number S.L. 31/2565.

**Institutional Review Board Statement:** Not applicable.

**Informed Consent Statement:** Not applicable.

**Data Availability Statement:** The data presented in this study, and any future updates, are openly available at https://dx.doi.org/10.21227/3q8d-jw96.

**Acknowledgments:** Gratitude is expressed for the generous support and funding received from the Metropolitan Waterworks Authority Thailand, particularly the Department of Resource and Environment, through their Real-Time Raw Water Quality Monitoring Station. Their contribution has played a vital role in the successful execution of our research. Sincere appreciation is extended to the Rajamangala University of Technology Thanyaburi for providing essential facilities that have greatly aided and facilitated our research efforts. Their assistance has been invaluable throughout the duration of this study.

**Conflicts of Interest:** The authors declare no conflict of interest.

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
