# Peer review of "Thailand Raw Water Quality Dataset Analysis and Evaluation"

_data, 2022_

Round 1
Reviewer 1 Report
The aim of this study was to describe a reconstructed daily water quality dataset that complements rare historical observations for six station points along the Chao Phraya River in Thailand using Internet of Things technology. The abstract does not show in detail the methodology used, i.e., how data analysis was carried out. Moreover, I suggest including numerical values in the abstract.
I suggest linking lines 75 and 76. One paragraph ends and the other begins without connection.
Lines 100 to 104: “The sensor measurements encompass parameters such as Turbidity (TURB_NTU), Optical Dissolved Oxygen (HDO), Dissolved Oxygen Saturation (DO_SAT), Spatial Conductivity (SPCOND), Acidity/Basicity (pH), Total Dissolved Solids (TDS), Salinity (SALINITY), Temperature (TEMP), Chlorophyll (CHL), and Depth (DEPTH).”
I suggest including the test method used (ASTM, ISO, Standard Methods, etc…)
Figures 1 and 2: Please, improve the quality of these figures.
Lines 150 and 151: “… which demonstrates a close correlation. Additionally, spcond exhibits a close relationship with both tds and salinity, indicating their interdependence.”
Please define “a close correlation” and “a close relationship”.
What were the % data set in the validation/calibration step and test step?
The authors used mean squared error (MSE) and root mean squared error (RMSE) as performance metrics. These metrics are perfect since the data behaviour is Gaussian. Please, comment in the manuscript about this assumption. On the other hand, I suggest including a reference about the cases in which the data behaviour departs from the normality, such as: Risk Assessment in Monitoring of Water Analysis of a Brazilian River. Molecules 2022, 27, 3628. https://doi.org/10.3390/molecules27113628
Minor editing of English language required.
Reviewer 2 Report
The Thailand Raw Water Quality Dataset describes a reconstructed daily water quality dataset that complements rare historical observations along six stations along the Chao Phraya River in Thailand. The data are original and well-defined, and the data collection methods are described in sufficient detail to be easily reproduced.
This paper employs Internet of Things (IoT) technology to collect data, the data set is technically sound. The data is in CSV format and can be easily reused by others.
The paper does not describe how to ensure the accuracy of the raw data collected, nor does it propose corresponding quality control measures. And the paper does not describe the later expansion of the dataset, for example, how to add more parameters to this dataset if they can be collected later.
The Thailand Raw Water Quality Dataset describes a reconstructed daily water quality dataset that complements rare historical observations along six stations along the Chao Phraya River in Thailand. The data are original and well-defined, and the data collection methods are described in sufficient detail to be easily reproduced.
This paper employs Internet of Things (IoT) technology to collect data, the data set is technically sound. The data is in CSV format and can be easily reused by others.
The paper does not describe how to ensure the accuracy of the raw data collected, nor does it propose corresponding quality control measures. And the paper does not describe the later expansion of the dataset, for example, how to add more parameters to this dataset if they can be collected later.
Round 2
Reviewer 1 Report
The authors addressed all of my comments.
Author Response
Thank you very much for your support and valuable comments.